# Respiratory syncytial virus ribonucleoproteins hijack microtubule Rab11 dependent transport for intracellular trafficking

Gina Cosentino[1], Katherine Marougka[1], Aurore Desquesnes[1], Nicolas Welti[1], Delphine Sitterlin[1], Elyanne Gault[2], Marie-Anne Rameix-Welti[2]*

**1** Université Paris-Saclay, Université de Versailles St. Quentin, UMR 1173 (2I), INSERM, Versailles, France, **2** Université Paris-Saclay, Université de Versailles St. Quentin; UMR 1173 (2I), INSERM; Assistance Publique des Hôpitaux de Paris, Hôpital Ambroise Paré, Laboratoire de Microbiologie, DMU15; Versailles, France

\* marie-anne.rameix-welti@uvsq.fr

**Data Availability Statement:** All relevant data are within the manuscript and its Supporting Information files. The nucleotide sequence of RSV-

## Abstract

Respiratory syncytial virus (RSV) is the primary cause of severe respiratory infection in infants worldwide. Replication of RSV genomic RNA occurs in cytoplasmic inclusions generating viral ribonucleoprotein complexes (vRNPs). vRNPs then reach assembly and budding sites at the plasma membrane. However, mechanisms ensuring vRNPs transportation are unknown. We generated a recombinant RSV harboring fluorescent RNPs allowing us to visualize moving vRNPs in living infected cells and developed an automated imaging pipeline to characterize the movements of vRNPs at a high throughput. Automatic tracking of vRNPs revealed that around 10% of the RNPs exhibit fast and directed motion compatible with transport along the microtubules. Visualization of vRNPs moving along labeled microtubules and restriction of their movements by microtubule depolymerization further support microtubules involvement in vRNPs trafficking. Approximately 30% of vRNPs colocalize with Rab11a protein, a marker of the endosome recycling (ER) pathway and we observed vRNPs and Rab11-labeled vesicles moving together. Transient inhibition of Rab11a expression significantly reduces vRNPs movements demonstrating Rab11 involvement in RNPs trafficking. Finally, Rab11a is specifically immunoprecipitated with vRNPs in infected cells suggesting an interaction between Rab11 and the vRNPs. Altogether, our results strongly suggest that RSV RNPs move on microtubules by hijacking the ER pathway.

## Author summary

Respiratory syncytial virus is the leading cause of severe lower respiratory infection in children worldwide and is increasingly recognized as a major respiratory pathogen in the elderly and frail. Yet, no curative treatment or vaccine is currently marketed. The late stages of RSV multiplication remain poorly understood despite they being potential targets for the development of antiviral strategies. In the infected cell, the viral genome is encapsidated and associated to the viral polymerase, to form the viral ribonucleoprotein

GFP-N was deposited in the Genbank nucleotide database with accession code OM326756. The dedicated PYTHON Script is available at GitHub (https://github.com/mawelti/RSV-RNP-TrackAnalysis).

**Funding:** MARW was supported by ATIP-AVENIR INSERM program (2018)(https://www.inserm.fr/nous-connaitre/programme-atip-avenir/), and the Fondation Del Duca - Institut de France (https://www.fondation-del-duca.fr/). GC and KM doctoral scholarships were supported by the Versailles St. Quentin university. The funders had no role in study design, data collection and analysis, decision to publish, or preparation of the manuscript.

**Competing interests:** The authors have declared that no competing interests exist.

(vRNP). The vRNPs are produced and assembled in cytoplasmic viral factories. The process ensuring their transport to the budding sites, at the plasma membrane, awaits to be precisely defined. Here we explored these mechanisms by tracking moving vRNPs in living infected cells. We developed an automated imaging pipeline allowing us to characterize the movements of vRNPs, at an unprecedented throughput. We then exploited the potential of our method to monitor the behaviour of the vRNPs during the infection. Using this approach, we document substantial trafficking of RSV along the microtubule network and demonstrate that RSV hijacks the recycling endosome pathway to promote the mobility of its vRNPs. Altogether, this work provides a cutting-edge approach allowing for live visualization of RSV RNP trafficking and critical data toward the understanding of RSV RNPs movements.

## Introduction

Respiratory syncytial virus (RSV) is the leading cause of severe lower respiratory tract infection in children worldwide. RSV infections are responsive for around 120 000 child death per year mostly in developing countries and are the main cause of child hospitalization in western countries [1, 2]. Periodic reinfections occur throughout life. Considered as benign in healthy children and adults, RSV infections are increasingly associated with significant morbidity and mortality in elderly and immunocompromised people with much the same disease burden as for influenza [3–5]. Despite the high burden of RSV infection, there is still no vaccine nor curative treatment available. The search for antiviral drugs is active, most of the candidates in development targeting entry steps and viral RNA synthesis [6].

RSV belongs to the *Mononegavirales* order. Its 15 kb negative sense single stranded genomic RNA encodes for 11 proteins. RSV is an enveloped virus exhibiting two major surface glycoproteins G and F. The matrix protein M coats the inner side of the viral membrane and surrounds the viral genomic material [7]. The genomic RNA is tightly encapsidated by the nucleoprotein N and forms a helical ribonucleocapsid. N is further bound to the polymerase (L) and its viral cofactors, the phosphoprotein P and M2-1, to form the viral ribonucleoprotein (vRNP) [8–10]. RNPs are the functional units driving viral RNA synthesis. Using the ribonucleocapsid as a template, L and P proteins ensure both the viral transcription and replication processes, while M2-1 is selectively required for the transcription process [9]. Viral RNA synthesis occurs within cytoplasmic inclusions, referred to as inclusion bodies (IBs), which can be regarded as viral factories [11]. The newly synthetized RNPs then reach the plasma membrane, where RSV presumably buds, forming elongated membrane filaments [12–14]. The matrix protein M is thought to drive virion assembly by bridging RNPs with the plasma membrane through its interactions with the RNPs and the cytoplasmic tail of the viral F glycoprotein [14–19]. The exact location of final virus assembly is debated since recent analysis of live G trafficking showed that RNP assemble with membrane glycoprotein prior to insertion into plasma membrane [20]. In any case, trafficking of RNPs to the cell surface requires active transport mechanisms, since diffusion of large objects in the cytoplasm is restricted by molecular crowding by organelles, the cytoskeleton and high protein concentrations [21]. This active transport might involve hijacking of the actin networks [22, 23] or of the recycling endosome (RE) pathway. RE is involved in delivering endocytosed material as well as cargos from the trans Golgi network to the plasma membrane [24]. Interestingly RE pathway has been implicated in RNP export of numerous viruses such as the influenza, Sendai, measles and mumps viruses [25–28]. The RE pathway is regulated by the small GTPase Rab11 which is present in three isoforms

(Rab11a, Rab11b and Rab25) in human cells. The GTP bound active form of Rab11 binds to target vesicle membranes thanks to its GPI anker and recruits specific factors ensuring transportation, docking or fusion of the vesicle to its cognate membrane [29]. Rab11 vesicles can traffic along both microtubules and actin networks by engaging various molecular motors by means of molecular adaptors called Rab11 Family Interacting Proteins (Rab11-FIPs).

Here we engineered a recombinant RSV expressing a fluorescent N to visualize the moving RNPs in living infected cells. An automatic quantitative analysis of RNP trajectories reveals rapid and directional motions that were abolished by nocodazole treatment consistent with transportation along the microtubule network. We observed a colocalization of approximately one third of the RNPs with Rab11a and RNPs moving together with Rab11a in infected cells. Inhibition of Rab11a expression impairs RNP movements. Interestingly RSV infection affects transferrin recycling, dependent of Rab11 pathway. Moreover, interaction between Rab11a and viral RNPs was confirmed by co-immunoprecipitation experiments. All of these data confirm that RSV is highjacking the Rab11-RE pathway to ensure RNPs export.

## Results

### Development of a recombinant RSV to monitor intracellular transport of RNPs

To investigate RNP dynamics in living cells, the RSV N protein was fused to a fluorescent tag to generate fluorescent RNPs. We previously demonstrated that a N-terminal fusion of the N protein does not affect the transcription-replication activities if co-expressed with a wild type N [11]. We thus engineered and rescued a recombinant RSV expressing both the wild type N and a GFP-N by inserting the GFP-N coding sequence flanked with transcription start and stop signals sequences between M and SH genes in our reverse genetic vector (S1A Fig) [30]. To ascertain whether the GFP-N would interfere with viral growth, we compared single cycle growth kinetics of wild type and N-GFP expressing RSV (hereinafter referred to as RSV-GFP-N) (S1B Fig). The RSV-GFP-N growth was similar to the wild type RSV and reached titers around $10^6$ PFU/mL in 24h, showing that the GFP-N addition did not significantly impair viral multiplication. We then investigated GFP-N protein localization in respect to the wild type N protein and known viral structures such as IBs and viral filaments. HEp-2 cells were infected by RSV-GFP-N for 24h and intracellular localization of RSV N, P and F proteins were determined by confocal imaging after immunostaining of each protein. The GFP signals were recorded in parallel. As previously described, IBs, cytoplasmic dots and viral filaments were observed by immunostaining of N and P, and F staining only revealed viral filaments (Fig 1) [31]. The GFP signals perfectly colocalize with N and P staining in IBs, viral filaments and small puncta in the cytoplasm. Of particular interest, we noted a perfect colocalization of GFP signals with both N and P staining in cytoplasmic dots which could be considered as RNPs (Figs 1 and S1C). Moreover, we observed colocalization of the GFP signals with F staining in viral filaments (Fig 1C, panel F). These data strongly suggest that GFP-N is incorporated together with the wild type N in the vRNPs. Altogether, the RSV-GFP-N was considered suitable as a novel tool marker to assess vRNP trafficking across the cytoplasm during infection.

### Dynamics of RNPs in RSV infected cells

To assess RNP dynamics in living cells, we analyzed HEp-2 or A549 cells infected with RSV-GFP-N for 18-20h by video-microscopy. High-rate time-lapse images were acquired under a confocal spinning disk microscope (S1 Movie). Numerous small GFP positive dots are

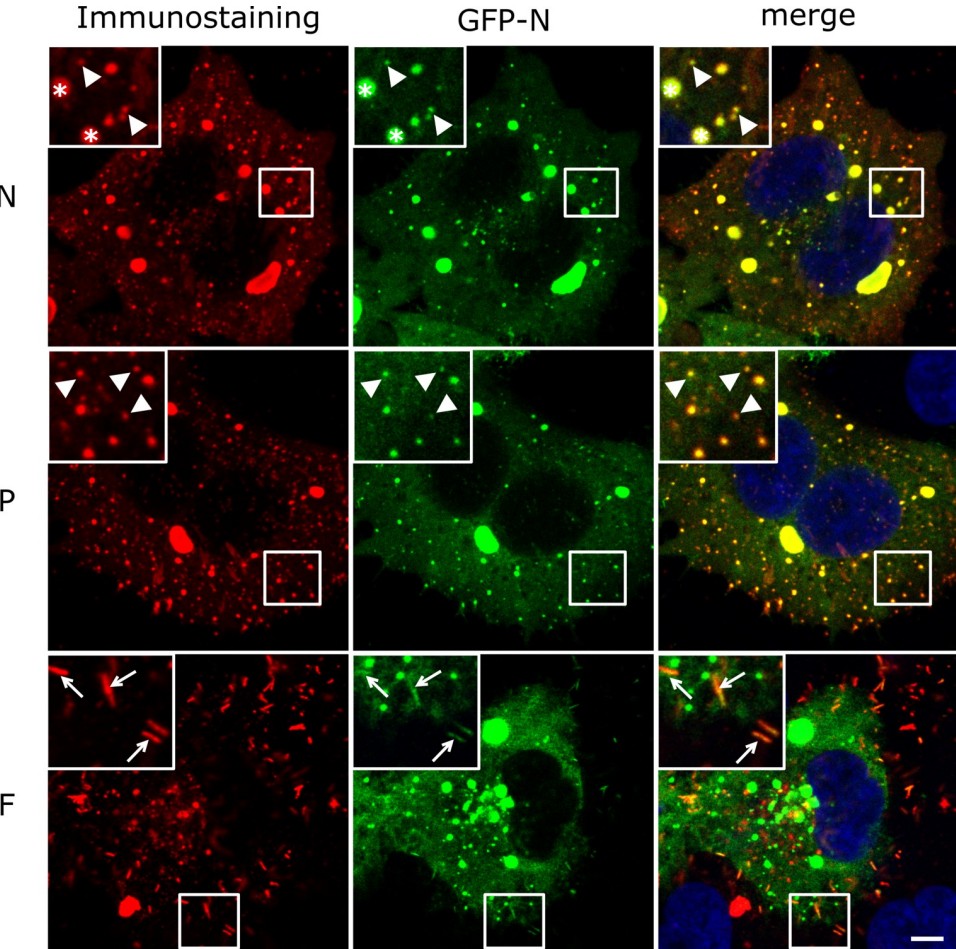

**Fig 1. Localization of GFP-N and wild type N, P and F proteins in RSV-GFP-N infected cells.** HEp-2 cells were infected with RSV-GFP-N. At 24h p.i. cells were stained with antibodies against N, P or F (red) and Hoechst 33342 (merge). The GFP-N protein was visualized through its spontaneous green fluorescence. IBs are indicated with a star, RNPs are indicated with white arrow heads and viral filaments are indicated with white arrows. Representative images from 2 independent experiments are shown. Images stacks (3 z-steps) were processed as maximum projections and visualized after gaussian filter fixed at 0.5. Scale bar 5 μm.

visible. Most dots exhibited slow motion movements with rapid and sudden changes of direction when some exhibited fast directed trajectories (white arrow head Fig 2A). Strikingly fast-moving particles appeared stretched (Fig 2A). Moreover, we observed some particles moving out of the IBs (arrowhead on S1 Movie). Automatic single particle tracking of the GFP positive dots was undertaken using Imaris software (S2 Movie) and resulted in hundreds of tracks per cells (266 to 2342, mean 993 from 23 cells in 4 independent experiments). Tracks located nearby IBs were removed from analysis as slight changes of fluorescence within IBs were prone to induce false-positive signals (S2A Fig). We also filtered tracks, in which particles instant speeds always remained below 50% of their maximal instant speed along the entire track recorded to remove artifactual links between the tracks of two slow moving objects (See S2B Fig and legend).

We then quantified several parameters characterizing intracellular movement of vRNPs. Displacement is characterized by 1) the *track length* which corresponds to the sum of lengths of displacement between two time points and 2) the *track displacement*, which corresponds to the minimum distance between the first and the last position of the particle (Fig 2B). *Track*

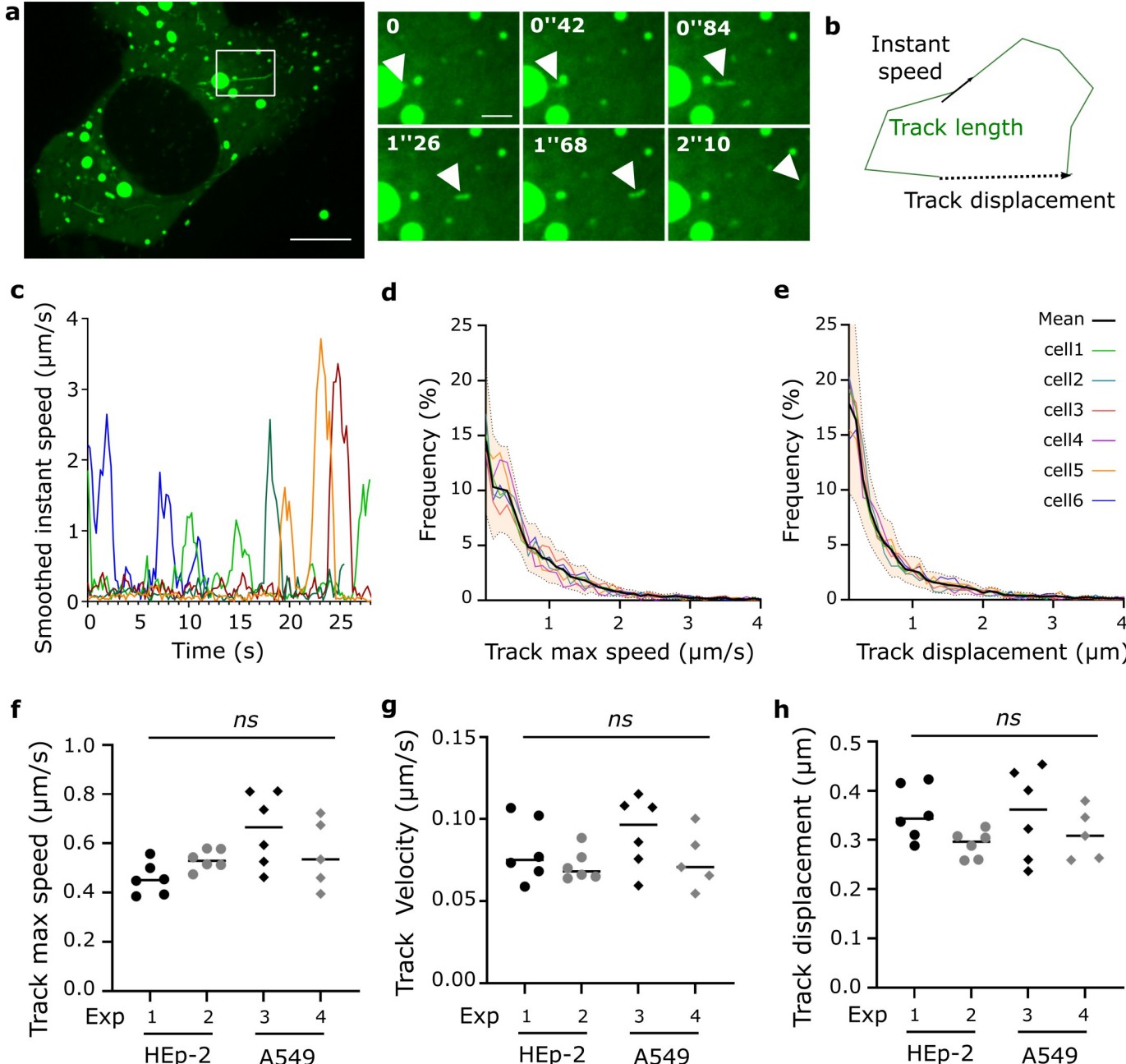

**Fig 2. Characterization of RNP intracellular motions in RSV-GFP-N infected cells.** Live images of HEp-2 or A549 cells infected with RSV-GFP-N for 18 to 20h are analyzed using Imaris software as described in methods section. **a)** Dynamic behavior of RSV RNPs in HEp-2 cells. Images stacks (3 z-steps) were processed as maximum projections and visualized after gaussian filter fixed at 0.5, the large view is a time projection of 12 consecutive frames (0.21 s between frames). Scale bar 10 μm. The zoomed time series show a fast-moving RNP pointed by white arrow heads, scale bar 2 μm. Representative images from 12 movies from 2 independent experiments are shown. **b)** Schematic illustration of the motion characterization parameters: instant speed (ratio between the minimal distance between two positions and time interval between these positions), track length (sum of the minimum distance between 2 consecutives positions for the whole track), track displacement (minimal distance between the first and the last position of the particle). **c)** Smoothed instant speeds of fast-moving particles plotted versus time from the first detection (each color represents one particle, representative examples). **d, e)** Cumulative distribution of track max speed and track displacement from 6 individual HEp-2 cells from one experiment. **f, g and h)** Each point is the median of track maximum speed, track velocity and track displacement from one individual cells. *ns*: No statistical difference between 4 independent experiments on HEp-2 and A549 cells using Brown-Forsythe and Welch's ANOVA test.

*velocity* is the ratio between track displacement and track duration. Track displacement and track velocity are thus expected to be high for particles exhibiting long range-oriented motion. *Smoothed instant speed* is the ratio between the minimum distance between the first and the fourth of 4 consecutive positions and the time interval between these positions (see S2F Fig and corresponding legend). Focusing on instant velocities of single fast-moving particles, we observed that rapid motions occur intermittently (Fig 2C). Thus, we also calculated the *track maximum speed* corresponding to the maximum smooth instant speed of the track. Cumulative distribution of track displacement for 6 cells in one experiment is shown in Fig 2E. Around 10% of the tracked particles exhibit a displacement over 1.8 μm when most of them exhibit small displacement (median ranging from 0.29 to 0.42μm). Likewise, cumulative analysis of particles maximal speeds shows that around 10% of the RNPs exhibit maximum velocities above 1.7μm/s (Fig 2D). The median value of each parameter reflects the overall characteristics of a particle's movements in an individual cell and provides a single value that can be used for further comparing treated groups. Median of track maximal speeds, track velocities and track displacements were not significantly different in 4 independent experiments on two different cell types (HEp-2 or A549). This points to a potential generalizability of vRNP's behavior during RSV infection in this model and indicates that these parameters are suitable for further group comparisons (Fig 2F–2H).

## RSV RNP rapid movements are dependent on microtubule network

Directional, rapid and discontinuous movements of the vRNPs are suggestive of microtubule (MT)-dependent transport [32]. We investigated vRNP localization relative to MT in HEp-2 and A549 cells infected with wild type RSV for 20h. N and MT were immunostained and cells were analysed on a super resolution confocal microscope (Airyscan, Leica). High-resolution images of infected cells revealed that MT were decorated with multiple vRNPs (Fig 3). To visualize the movements of vRNPs relative to MT, RSV-GFP-N infected A549 cells were treated with docetaxel–AF 647, a fluorescent-linked drug binding MT polymer. This enabled MT live staining and dual colored time-lapse imaging acquisition. Fluorescent vRNPs moving rapidly along microtubial structures were clearly visible demonstrating RNP transport along MT network (S3 Movie and Fig 4A). Acquisition in two color channels prevented any quantitative analysis of vRNP trajectories. To further dissect the involvement of MT network in RNP movements, we analyzed the effects of MT depolymerisation on RNPs movements. Living RSV-GFP-N infected cells were imaged before and after a 10 min treatment with nocodazole (20μM). Importantly, our protocol allowed imaging the same cells before and after drug-treatment. Remarkably, the nocodazole treatment abolished almost all the rapid and directed movements (S4 Movie). In contrast, cells treated with Cytochalasin D, a drug impairing actin polymerization, still displayed fast moving vRNPs (S5 Movie). Finally, we applied our pipeline of GFP particles tracking on mock or nocodazole treated cells, and observed a significant decrease of track displacement, track velocity and track maximum speed (Fig 4D–4F, S6 and S7 Movies). A centered projection of the tracks of one cell during 60 s is shown in Fig 4 to illustrate the strong reduction of track displacement in nocodazole treated cells. Altogether, these data revealed that fast-directed movement of RNPs rely on MT network. Consistent with the role of MT in RNP transportation, nocodazole treatment of RSV infected cells after entry steps significantly decrease progeny virus production as previously reported (S3 Fig) [33].

## Rab11 and RNPs colocalize in infected cells

Recycling endosome vesicles, marked by the small GTPase Rab11, are known to be involved in RNP trafficking of several negative strand RNA viruses [25–28]. To investigate the recycling

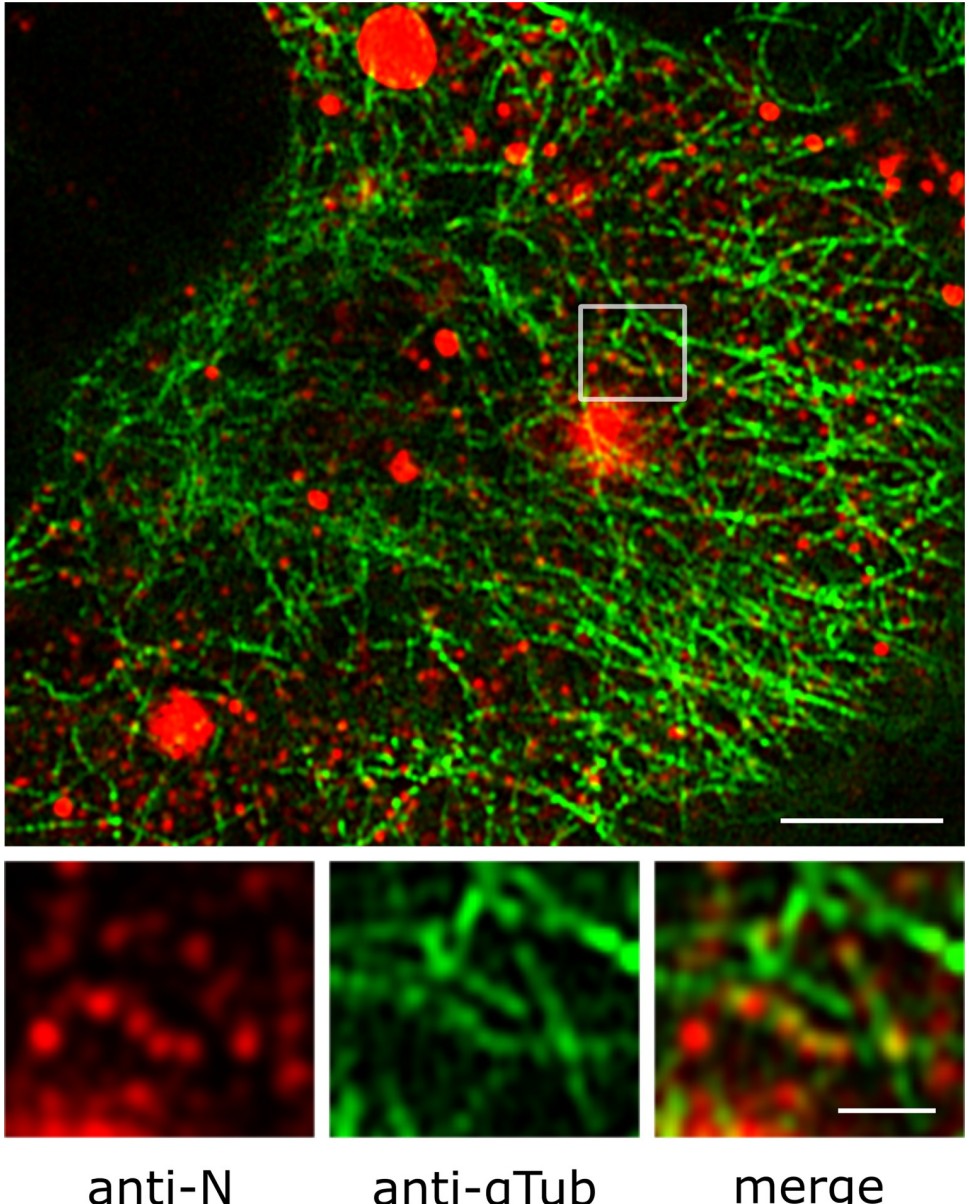

**Fig 3. RNPs are decorating the MT network.** A549 cells were infected with RSV for 20h. N (in red) and α-Tubulin (in green) are revealed by immunostaining. High-resolution images were generated using a confocal microscope with Airyscan detector. Images were visualized after gaussian filter fixed at 0.5. A representative image of a whole cell out of 5 cells in 3 experiments is shown. RNPs decorating the MT network are shown on the zoom area. Scale bars 5μm and 1 μm.

endosomes involvement in RSV RNPs trafficking, we analyzed Rab11a and RSV RNP intracellular localizations. We infected A549 cells constitutively expressing HA-Rab11a (A549-HA-Rab11a) with wild type RSV [34]. At the indicated time points, cells were fixed and N and HA-Rab11a were detected by immunostaining. Both N and HA staining revealed numerous individual cytoplasmic dots corresponding respectively to vRNPs and Rab11a positive endosomes, which exhibited partial colocalization (Fig 5). To quantify the colocalization between Rab11a and N positive dots, we extracted spot detections and performed object-based

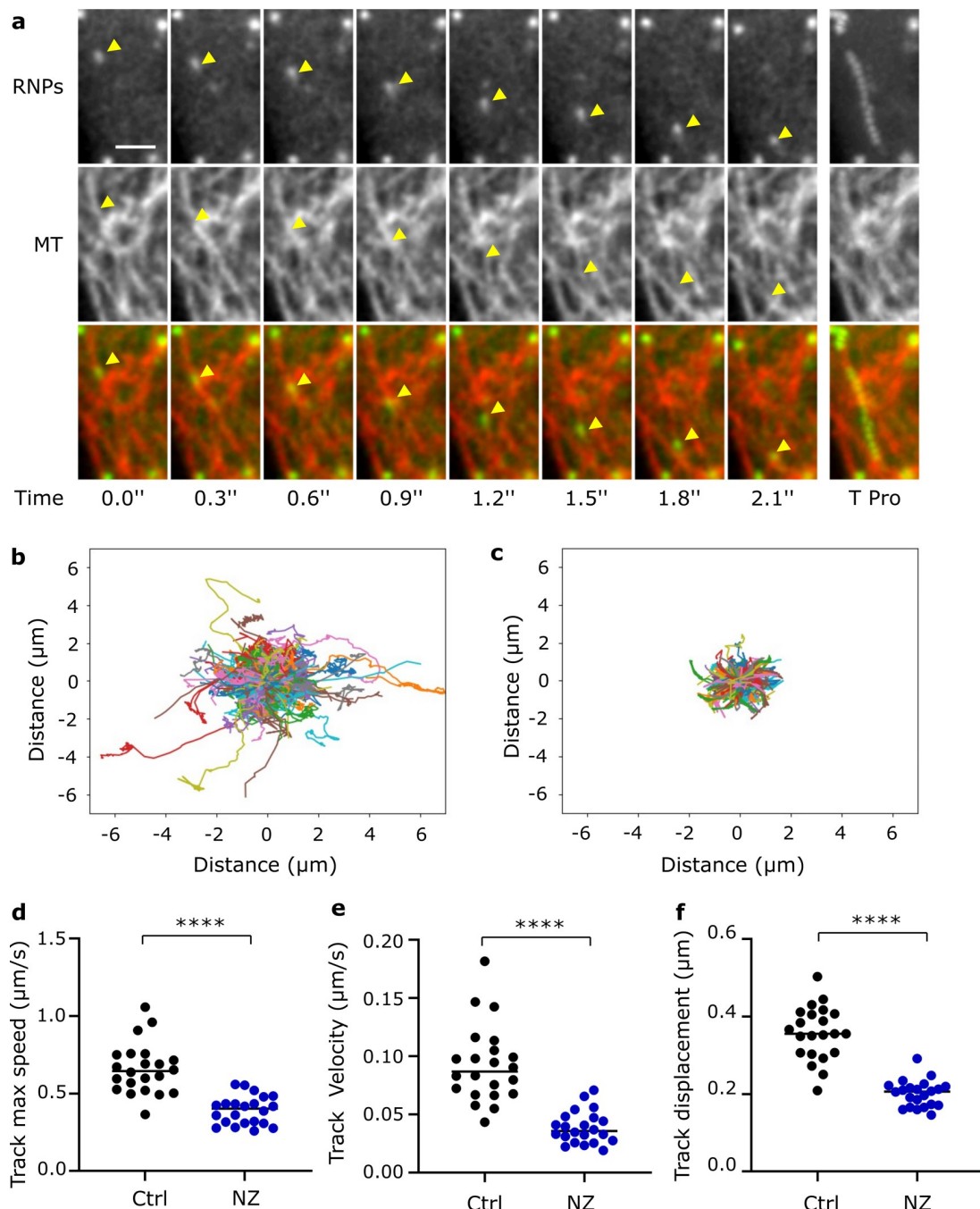

**Fig 4. Fast long range RNP motions are dependent on microtubules network. a)** Time series of live images of RNPs (in green) moving along MT stained by fluorescent docetaxel-647 (in red) in RSV-GFP-N infected A549 cells. Images stacks (2 z-steps) were processed as maximum projections and visualized after gaussian filter fixed at 0.5. Yellow arrows point positions of a moving RNP. The last image shows a time projection (T Pro). Scale bar 2µm. **b to f)** RSV-GFP-N infected HEp-2 cells were treated at 17 h.p.i. with nocodazole (10µM, NZ) or DMSO (Ctrl) for 1 h before live-imaging and track analysis. **b, c)** Centered projection of the tracks of RNPs in a mock (c) or a nocodazole (d) treated cell analyzed over 60 s. Representative images. **d, e, f)** Each data point represents the median of track maximum speed, track velocity and track displacement from one individual cell. **** *p* < 0.0001 using t test with Welch's correction. Data are from 22 cells (DMSO) and 23 cells (NZ) from 3 independent experiments.

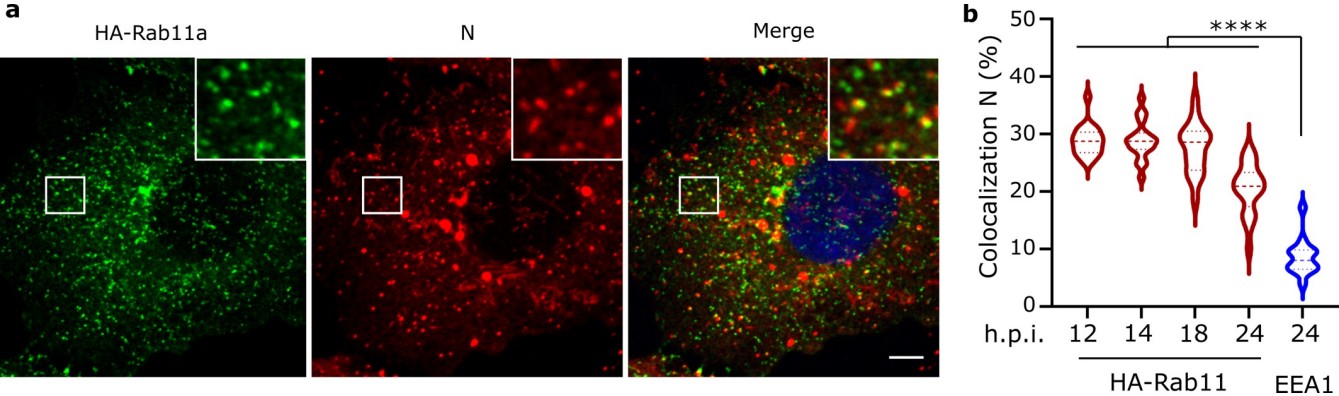

**Fig 5. Colocalization of RNPs and Rab11a in RSV infected cells. a)** A549-HA-Rab11a cells infected with RSV for 18 h. Rab11a (green) and RNPs (Red) were detected by immunostaining of HA and N. Images were visualized after gaussian filter fixed at 0.5. Scale bar 5 μm. The boxed areas enclose Rab11a and N spots shown magnified. **b)** Percentage of N spots colocalizing with HA-Rab11a or EEA1 positive spots calculated using Icy Software. Violin plots show the median (center line) and the first and third quartiles (upper and lower hinges). Results are from 70 cells from 3 independent experiments. **** $p < 0.0001$ using Brown-Forsythe and Welch's ANOVA test followed by Dunn's test for multiple comparisons.

colocalization analysis using Icy software. This method first segments the spots and then analyzes their relative distribution with second order statistics enabling robust quantification of colocalized objects [35]. This analysis revealed that approximately 30% of N spots colocalize with Rab11a positive vesicles at 12 to 18h p.i. with a slight decrease at 24h p.i.. In contrast, less than 10% of N spots colocalize with EEA1, a marker of early endosomes used as a negative control. Similar results were obtained in A549 and HEp-2 cells (S4 Fig). These data suggest an involvement of Rab11a in vRNP traffic.

## Rab11 pathway is involved in RNP fast and directed movements

To determine if Rab11a could be involved in RNP trafficking, we set up an experiment to visualize both RNP and Rab11 movements in living infected cells. A549 and HEp-2 cells were infected with RSV-GFP-N and transiently transfected with a mCherry-Rab11a expression vector. At 18-20h.p.i., we performed dual color time-lapse imaging to assess the dynamic of both Rab11a and RNPs. Interestingly, we observed fast-moving RNPs moving together with Rab11a positive vesicles demonstrating that at least some of the fast-moving RNPs are associated with Rab11a positive structures (Fig 6A and S8 Movie). Strikingly, RNPs appeared as pulled by the Rab11 positive spots. RNPs associated with Rab11 positive vesicles were no longer mobile when cells were treated with nocodazole suggesting that these movements were microtubule-dependent (S9 Movie). The dual color acquisition settings unfortunately prevent automatic tracking of RNPs and Rab11a in this experiment.

To further demonstrate the essential role of Rab11 in RNP export, we performed SiRNA depletion experiments. A549 cells were transiently transfected with small interfering RNA (SiRNA) targeting Rab11a or control scrambled SiRNA for 48h and subsequently infected with RSV-GFP-N for 18-20h before live imaging. Western blot analysis of the whole cell lysates using an antibody against Rab11a showed a considerable decrease in Rab11a levels in SiRNA targeting Rab11a treated cells compared to the control (S4C Fig). RNP movements were quantified as described above. The track velocity, track maximum velocity and track displacement were significantly reduced in Rab11a silenced cells as compared to the control (Fig 6 and S10 and S11 Movies). The centered projection of the tracks from one cell illustrates the reduction of track displacement. These results suggest that Rab11a is involved in RNP trafficking.

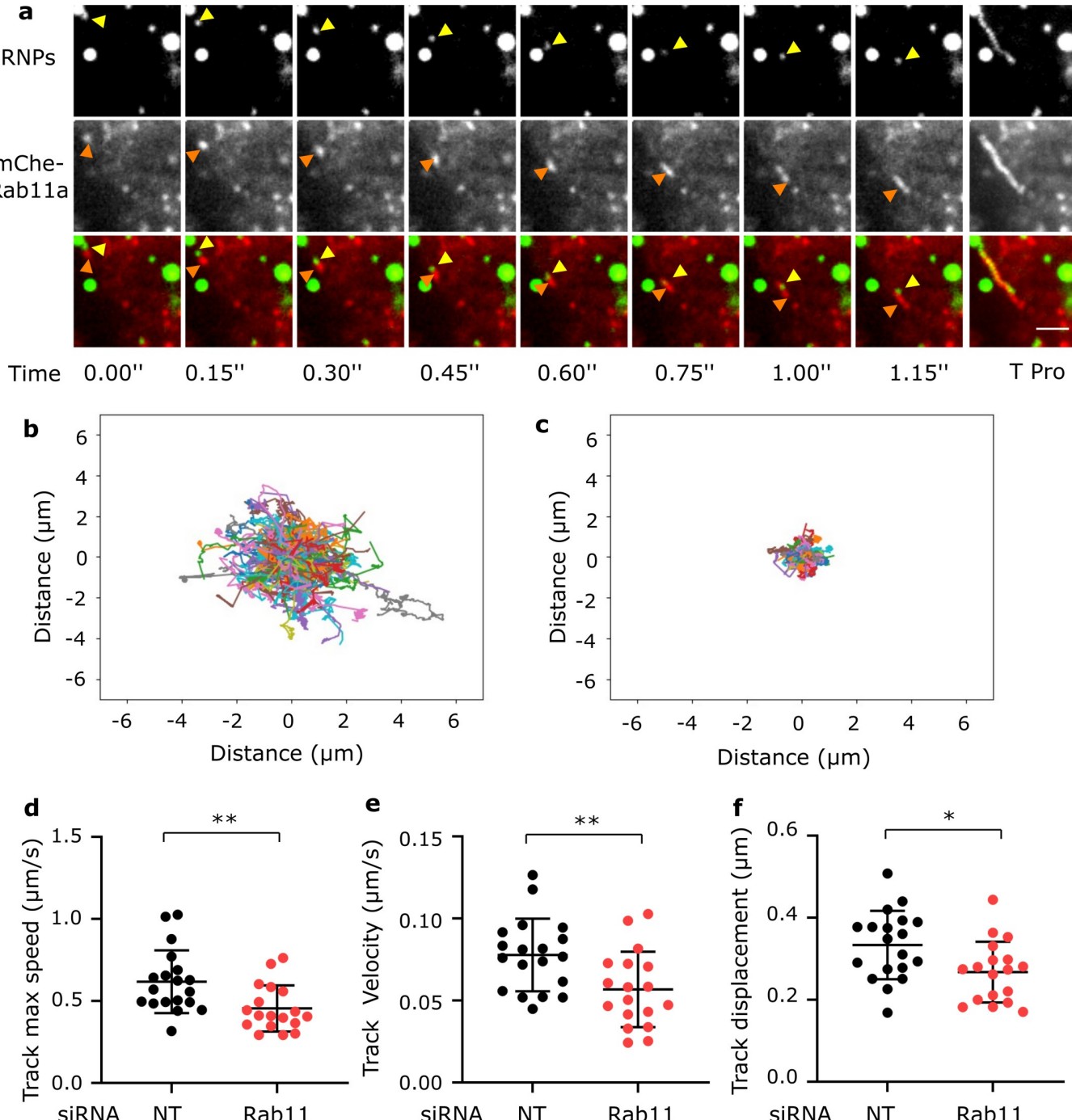

**Fig 6. Rab11 is involved in fast and directed RNP movements. a)** Time series of live images of A549 cells transiently expressing mCherry-Rab11a (red) and infected for 18h with RSV-GFP-N (in green). Images were visualized after gaussian filter fixed at 0.5. Positions of a RNP moving together with a mCherry-Rab11a positive object are pointed out by yellow (RNP) and orange (Rab11a) arrow heads. The last image shows a time projection (T Pro). Scale bar 2μm. **b to f)** A549 cells were transiently transfected with siRNA targeting Rab11a or non targeting ones (NT) for 48h then infected with RSV-GFP-N for 18 h before live-imaging and track analysis. **b, c)** Centered projection of the tracks of RNPs in a NT (c) or Rab11a (d) KO cells analyzed during 60 s. Representative images. **d, e and f)** Each data point represents the median of track maximum speed, track velocity and track displacement from one individual cell. ** $p < 0.01$, * $p < 0.05$ using t test with Welch's correction. Data are from 19 cells (NT) and 18 cells (Rab11a) from 3 independent experiments.

Since our data indicate that RSV hijacks the Rab11a pathway for RNP export, we examined whether RSV infection impacted in turn RE function. To this end, we explored transferrin recycling, which is dependent on RE functioning. Mock or RSV infected A549 cells were incubated in the presence of fluorescent transferrin and were then placed in fresh medium. Cytoplasmic fluorescence (corresponding to non-recycled transferrin) was quantified after 0 and 20 min of incubation in fresh medium (S5 Fig). Interestingly, the infected cells exhibit an increase in transferrin recycling as compared to uninfected ones suggesting an acceleration of the traffic of RE. Altogether, these data suggest that RSV not only uses the recycling endosome pathway but also manipulates its functioning.

## Biochemical validation of RNP-Rab11a association

We further assessed the association between RNPs and Rab11a performing co-immunoprecipitation (IP) analysis. A549 cells stably expressing HA-Rab11a were infected for 16h with RSV-GFP-N or RSV expressing GFP (RSV-GFP) as a control. Cell lysates were incubated with beads coated with anti-GFP antibodies and IP was carried out. Both pre-purification and IP samples were analyzed by western blot to reveal N, P and HA-Rab11 (Fig 7A). Wild type N and P proteins were efficiently purified together with the GFP-N protein suggesting that the whole RNP is co-immunoprecipitated with GFP-N. The HA-Rab11a protein was also repeatedly co-immunoprecipitated with the GFP-N when no corresponding band was visible in the control. Consistent with the colocalization and trafficking experiments, only a small fraction of Rab11a was co-immunoprecipitated with the RNPs (Fig 7A, bound fraction). The mirror IP was performed on A549-HA-Rab11 cells infected with wild type RSV for 16h using anti-HA antibodies. N protein was found in the IP fraction of Rab11HA expressing cells but not of wild type A549 cells used as control (Fig 7B). Interestingly P was also co-immunoprecipitated with HA-Rab11a suggesting that Rab11 is interacting with the RNP. To further investigate this point, we performed similar IP experiments on cells infected with a recombinant RSV expressing a L protein fused to GFP (RSV-L-GFP). The GFP is inserted in the hinge 2 region of L [11]. Western blot analysis of the IP fraction failed to reveal the GFP-L most likely due to a lack of sensitivity and poor transfer of a large protein like L (Fig 7A; RSV-L-GFP). However, N and P were co-immunoprecipitated thus validating the IP and confirming the capture of the

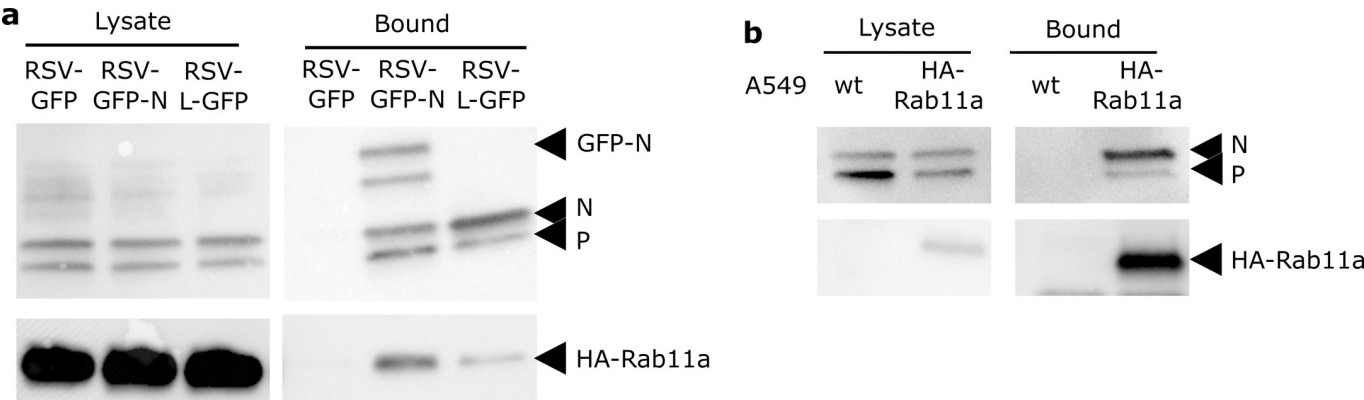

**Fig 7. Co-immunoprecipitation of Rab11a and RSV RNPs in infected cells.** (a) A549 stably expressing HA-Rab11a were infected with RSV-GFP, RSV-GFP-N or RSV-L-GFP as indicated for 16h, lysed and incubated with beads against GFP. Cell Lysates and bound fractions were analyzed by western-blot using rabbit polyclonal anti-N, anti-P and anti-HA antibodies. One representative experiment out of 3 (RSV-GFP-N) or 2 (RSV-L-GFP) is shown. (b) A549 WT or stably expressing HA-Rab11a as indicated were infected with RSV for 16h, lysed and incubated with beads against HA. Western blot analysis of the co-immunoprecipitated proteins is shown. One representative experiment out of 2.

RNP together with the GFP-L. Finally, we detected HA-Rab11a repeatedly and specifically in the IP sample consistent with an interaction between Rab11a and the RNP.

## Discussion

The understanding of many aspects of RSV multiplication has progressed considerably in recent years. However, the late stages of the viral cycle remain poorly understood. In particular, the mechanisms of transport of RSV RNPs from cytoplasmic viral factories to viral assembly sites at or near the plasma membrane remain unexplored. In this study we were able to observe for the first time the movement of RNPs in infected cells using a recombinant RSV. Our strategy was to engineer a recombinant virus expressing an N protein fused to a fluorescent tag to produce fluorescent RNPs. To our knowledge this is the first description of a RSV expressing a tagged N. Expression of this additional GFP-N does not appreciably alter the replicative capacities of the virus. Noteworthy, fluorescent N perfectly colocalizes with the wild type N in the viral structures (IBs, RNPs and viral filaments) strongly suggesting that GFP-N is indeed incorporated in RSV RNPs. These data validate the use of the RSV-GFP-N as a new tool to study trafficking of RSV RNPs. In theory, more than 2000 N monomers are required to encapsidate the RSV 15kb genomic RNA [10]. GFP-N is less abundant than the wild-type N (Fig 7A) which might result from the transcription gradient. Nonetheless, the oligomerization of N and GFP-N on the viral RNA leads to an amplification of the fluorescent signal and allows for the detection of vRNPs. The GFP-N virus thus represents a tool of choice for the monitoring of vRNPs in living cells, although we can not rule out that GFP-N incorporation into the RNPs impairs their movements. An alternative strategy based on the labeling of the viral genome with nucleotide probes has been used to analyze the dynamics of RNPs during entry or the movement of viral filaments at the cell surface. However, this method did not enable RNP trafficking analysis [12, 13]. vRNPs are small, highly mobile objects. To track them, we had to find a compromise between the number of optical slices (z) analyzed and the time interval between 2 images. Our optimized pipeline relies on the acquisition of images on 2 to 3 optical planes with a total thickness of 0.7 to 2.3 μm and time-frame intervals of 0.07 to 0.21s. Our experiments were conducted on HEp-2 or A549 cells which are very thin as compared to some other cell types. Expanding the method to other cells is likely to require further optimization. In these cells, these parameters allowed a proper automatic tracking of mobile RNPs that are visible in the analyzed volume on several successive images. The potential of our method allowed us to generate high-throughput live imaging datasets otherwise inaccessible with more classical approaches. We were able to track hundreds of particles per cell and provide a robust characterization of vRNP movements in different conditions. A large proportion of the observed vRNPs appeared immobile or animated by low amplitude "Brownian movements". The resolution of our microscopy methods (around 250nm) does not allow fine characterization of very low amplitude non-directional movements. In contrast, during the 60 s acquisitions we performed, we observed that about 10% of the vRNPs were animated by fast and directional movements." These are believed to be the ones actively trafficking. The characteristics of these motions, in particular the maximum speed above 1–2 μm/s are in themselves suggestive of MT-related transport [32].

The importance of the microtubule network in RSV multiplication has already been established by previous studies. Treatment of infected cells with nocodazole was shown to reduce virion production by about 1 log with no indication of which stage of multiplication was disrupted [22, 33]. Here, our results clearly establish that vRNPs are transported rapidly and over long distances on the MT network. Indeed, we observed vRNPs moving on MTs and were able to verify the almost complete disappearance of these movements upon disassembly of the MT

network. Many negative strand RNA viruses also hijack the MT network and its associated motors for the transport of their RNPs. For example, RNPs from Influenza A, Sendai or measles viruses move along this network [25, 26, 36–38]. As what has been observed for RSV, depolymerization of the MT network does not completely abolish the replication of these viruses [25, 26, 39]. For influenza viruses, conflicting results were reported on the effect of nocodazole treatment on viral multiplication [40] or on movement of vRNPs [36]. These data suggest that alternate pathways exist for the transport and/or spread of these viruses. Relative contribution of MT transportation into RSV assembly is yet to be determined. It is worth noting that our study, like other published studies, was performed on epithelial cell lines, and not on differentiated polarized cells, which are the *in vivo* target cells of these viruses. In such cells cytoskeleton organization is different and the extent of MT dependent transportation and alternative transport pathways remains to be explored. Cortical F actin network has been involved in RSV filaments assembly and budding [23, 33, 41, 42], this might suggest that RNPs are transported from the cytoplasmic IBs near the plasma membrane thanks to MT network before interacting with F actin partners to promote virion assembly and budding.

Our data clearly demonstrate that the fast and long-distance transport of RSV RNPs is dependent on the recycling endosome. Inhibition of Rab11a expression indeed significantly reduces the travel length and the speed of vRNPs. However, vRNP transport is not completely suspended, suggesting either a partial inhibition of Rab11a expression, an involvement of Rab11b and/or the existence of alternative transport pathways. Direct observation of joint movements of Rab11a with vRNPs in living cells clearly demonstrate the involvement of RE in vRNP transport. Our videos show vRNPs literally pulled by the Rab11a positive structures. The immunoprecipitation experiments confirm an association between vRNPs and Rab11a although not demonstrative of a direct interaction. At a given time point about 30% of vRNPs are colocalized with Rab11a suggesting that the association of vRNPs with Rab11a is transient. In line with this observation, only a fraction of Rab11a and vRNPs are co-immunoprecipitated and only a small proportion of vRNPs are engaged in fast movement. Rab11 is able to engage motors linked to either microtubule or actin networks. The movements of Rab11a-associated vRNPs we observed, are fast and abolished by MT depolymerization demonstrating that they are MT dependent. Overall, our data support the following model for RSV RNP export: 1) vRNPs transiently associate with Rab11a-positive vesicles 2) these Rab11a-positive vesicles transport vRNPs rapidly and over long distances along the microtubule network. Moreover, RSV not only diverts the Rab11 pathway for its RNP transport but also manipulates it. Indeed, we observed that RSV infection increases transferrin recycling, consistent with previous analysis of Rab11a vesicles motion that revealed increase in velocity and in high distance displacement in RSV infected cells [37].

In recent years Rab11 has emerged as a key cellular factor involved in various stages of the multiplication of several viruses. Some viruses enter the cell using the RE compartment as suggested by the association of HHV8 or dengue viral capsids with Rab11 endosomes soon after infection [43, 44]. Furthermore, it has previously been shown that effectors of the Rab11 pathway: FIP1, FIP2 and Myosin Vb support RSV release at the apical side of the membrane. FIP2 is even thought to regulate the length of viral filaments [45, 46] suggesting an involvement of the RE compartment in the budding steps of RSV. Rab11 and its partner Rab11-FIP3 have also been implicated in the budding of influenza viruses [47] and recent results show that the integrity of the Rab11 pathway is required for ebola virus VP40 transfer to the membrane and for VLP release [48]. Subversion of Rab11 pathway for the transport of the newly synthetized RNPs of negative RNA viruses to plasma membrane is extensively documented. Previous studies showed that newly synthesized RNPs from measles, mumps, influenza, parainfluenza 1, Sendai viruses or a new world hantavirus are associated with Rab11 positive endosomes and

that Rab11 is involved in their transfer to the plasma membrane [25–28, 38, 39, 47, 49–51]. However, the precise mechanisms of vRNP associations with Rab11 and the pathways involved in transport remain to be defined. The numerous studies of the export of influenza virus RNPs have shown that vRNPs recruitment depends on a direct protein-protein interaction between a unit of the viral polymerase complex PB2 and the active form of Rab11a, and that the Kinesin Kif13a is one of the motors involved in their transport to the membrane [38, 39, 52]. The L polymerase of Sendai virus has also been shown to be involved in the recruitment of vRNPs to RE vesicles [53]. Rab11a is co-immunoprecipitated with RSV L suggesting that the RSV poly-merase could similarly promote Rab11a-vRNP association. However, since the whole vRNP is captured in our experiments, further investigations are needed to determine if L is critical for recruiting Rab11.

In conclusion, our results establish that RSV RNPs associated with Rab11a vesicles are transported along MTs. These data are a first step forward in the understanding of vRNP export. They pave the way for further work aimed at identifying the viral proteins, the Rab11 cellular partners and the molecular motors involved in vRNP transport. In the future, interactions between RSV and the ER pathway could represent a new therapeutic target to block the late stages of the RSV replication cycle.

## Methods

### Cells

HEp-2 cells (ATCC, CCL-23) and A549 cells (ATCC, CCL-185) were maintained in Eagle's minimum essential medium (MEM) and in Dulbecco modified essential medium (DMEM), respectively, supplemented with 10% heat-inactivated fetal bovine serum (FBS) supplemented with penicillin–streptomycin solution. A549 cells constitutively expressing the HA-Rab11a (WT) are a kind gift from Dr N Naffakh [34] and were grown in DMEM supplemented with 1μg/mL puromycin. Cells were grown in an incubator at 37˚C in 5% $CO_2$. HEp-2 cells and A549 tested negative for mycoplasma using a MycoAlert PLUS Mycoplasma Detection Kit (Lonza).

### Virus and plasmids

All the viral sequences were derived from the human RSV strain Long, ATCC VR-26. The wild-type RSV, the RSV-GFP, the RSV-GFP-N and the RSV-L-GFP were rescued by reverse genetics as previously described [30, 54]. To construct the reverse genetic vector for RSV-GFP-N, the GFP-N coding sequence was amplified from the pmGFP-N vector [11] by PCR using specific primers (sequence available upon request) containing RSV gene start and gene end sequences and was cloned into a Eco105I restriction site between the RSV M and SH genes in the pACNR-rHRSV [30]. The nucleotide sequence of RSV-GFP-N was deposited in the Genbank nucleotide database with accession code OM326756. pmCherry-Rab11a was con-structed by inserting the Cherry coding sequence in peGFP-Rab11a [55] (kind gift from Dr Sauvonnet) between NheI and XhoI restriction sites. All constructs were verified by sequenc-ing. Experiments were performed with viral stock amplified on HEp-2 cells at 37˚C after three to five passages. Plaque assay were performed at 37˚C on HEp-2 cells using Avicel overlay as previously described [30].

### Antibodies and reagents

The rabbit polyclonal anti-P and anti-N were obtained by repeated injection of purified recombinant protein produced in *Escherichia coli* as previously described [56]. The rabbit

anti-Rab11a is from Thermofisher, the rabbit anti-EEA1 and rabbit anti-HA from Cell signaling, the rat anti-HA is from Roche, and the mouse anti-F was from Abcam. The mouse monoclonal anti-P 021/2P [57] is a kind gift from Dr JF Eléouët. Secondary antibodies raised against mouse or rabbit IgG (H+L) and conjugated to Alexa Fluor 488, 594 or 647 were from Thermofisher. Secondary antibodies raised against mouse or rabbit IgG (H+L) and conjugated to horseradish peroxidase were from Promega. DMSO, Nocodazole and Cytochalasin D were from Sigma. Live staining of Microtubules was achieved with SiR-tubulin Kit 647 as described by the manufacturer (Spirochrome).

## Cell infection and transfection

All infections were performed at 37˚C on exponentially growing HEp-2 or A549 cells at a multiplicity of infection of 1 (except indicated otherwise). For Rab11mCherry and GFP-N live imaging, A549 or HEp-2 cells were seeded at 90% confluence in Ibidi μ-Slide Angiogenesis (Ibidi) and were transfected with 0.5μg of pmCherry-Rab11a (per 0.5 $10^6$ cells) using Lipofectamine 2000 (Invitrogen) according to the manufacturer recommendations. Then cells were infected with RSV-GFP-N 1h post-transfection. SiRNA transfections were performed on A549 cells newly seeded on Ibidi μ-Slide Angiogenesis. Briefly, a pool of 4 different siRNA targeting the same gene Rab11a (ON-TARGETplus SMARTpool, Dharmacon) or a non targeting pool was transfected simultaneously to the cell seeding, using LipoRNAiMax (Thermofisher) transfection reagent. 0.3 μL of LipoRNAiMax per well (per $0.2.10^5$ cells) were used with a final siRNA concentration of 10 nM. Cells were incubated at 37˚C for 48h before being infected.

## Immunofluorescence–Colocalization analysis

Cells were grown on glass coverslips in 24 wells plates. After infection at high MOI for the indicated times, cells were fixed with PBS-formaldehyde 4% (v/v) for 10 min, washed with PBS and permeabilized with PBS-BSA 1% (w/v)-Triton X-100 0.25% (v/v) for 10 min. Cells were incubated for 1 h in PBS-BSA 1% (w/v) with the appropriate primary antibodies, rinsed with PBS then incubated for 30 min with the appropriate Alexa Fluor-conjugated secondary antibodies and with Hoechst 33342 (1 μg/ml). Coverslips were rinsed in PBS, then mounted in ProLong diamond antifade reagent (Thermofisher). Cells were examined by confocal microscopy under a WLL Leica SP8 microscope except indicated otherwise. Representative pictures were taken.

Colocalization analysis were performed using the open-source software Icy (http://icy. bioimageanalysis.org/) [58]. We used Spot Detector plugin to automatically detect objects corresponding to RNP and vesicles (Rab11a or EEA1) and then Colocalization Studio plugin method: object based and statistical colocalization, SODA [35], to quantify the number of RNP at a distance less than 270 nm from the cellular vesicles.

## Transferrin uptake quantification

A549 cells starved for 30 min in serum-free medium containing 0.5% (w/v) BSA were pulsed for 10 min with Tf-AF-647 (25 μg/mL, Thermofisher). Cells were then quickly incubated with cold acid buffer (0,5 M Glycine, pH 2.2) to dissociate surface-bound Tf, extensively washed with PBS, and incubated at 37˚C in complete media for 0 or 20 min before fixation with 4% paraformaldehyde. Cells were imaged by confocal microscopy under a WLL Leica SP8 microscope. Cytoplasmic AF647 fluorescence signal was quantified using the "Measure" function of imageJ/Fiji on average z-projections. Signal intensities at 20 min were normalized to their respective mean at 0 min.

## Co-Immunoprecipitation

WT or HA-Rab11a expressing A549 cells infected with RSV-GFP-N, RSV-L-GFP, RSV-GFP or RSV-WT for 16 hours were lysed in a co-IP lysis buffer (Tris 25mM pH 7.2; NaCl 150 mM; IGEPAL CA-630 0.2% (Sigma); glycerol 10% (v/v); EDTA 0.5mM; antiprotease and phosphatase (Thermofisher)), then incubated overnight at 4˚C with GFP-Trap beads (Chromotek) or Magnetic-HA beads (Thermofisher) according to the manufacturer's recommendations. Beads were rinsed three times in a co-IP dilution buffer (Tris 25mM pH 7.2; NaCl 150mM), then eluted in Laemmli buffer at 95˚C and analyzed by SDS-PAGE.

## Live Imaging

Live imaging was performed on A549 or HEp-2 cells seeded on Ibidi µ-Slide Angiogenesis, infected and treated as described above. One-minute time-lapse acquisitions were performed at 37˚C and 5% $CO_2$ using an inverted confocal microscope with a 100x oil-immersion lens (Plan-APOCHROMAT), a CSU-X1 spinning-disk head (Yokogawa, Japan) and a sCMOS-PRIME 95B (Photometrics) camera. The whole setup was driven with MetaMorph software (Molecular Devices, Sunnyvale, CA, USA). One to three z sections with a step of 0.8 µm were acquired at 70 to 210 ms intervals for 1 min. Laser intensity was set between 10–20% power, and acquisition time was 50 ms. The raw data were processed using ImageJ software (National Institutes of Health, Bethesda, MD, USA) to perform maximum projections of z stacks images and background subtraction. Tracks analysis was performed on Imaris software 9.5.0 (Bitplane Inc.). Briefly fluorescent spots were identified using Imaris built-in spot generation algorithm (with a seeding spot size of 250 nm). Spot generation was achieved manually for each dataset using intensity at the center of each spot as the threshold. Quality threshold was fixed above 20.0. The spots were tracked over time to generate motion statistics for each cell using autoregressive motion algorithm (max distance 1µm, max gap size 2). Imaris output file "Position. csv" was further used to calculate the characteristic of each track (*track duration*, *track length*, *track displacement*, *track maximum speed*, *mean track velocity*) and to filter tracks using dedicated PYTHON Script available at GitHub (https://github.com/mawelti/ RSV-RNP-TrackAnalysis). All tracks smaller than 4 steps were filtered. Coordinates of the IBs area were entered manually to remove false tracks generated by frequent false detections due to high background signal in the vicinity of the IBs (see S2A and S2B Fig). Tracks whose instant speed is always lower than 50% of the maximum instant speed were also filtered to remove false link between 2 slow moving objects (See S2C–S2E Fig). *Smoothed instant speeds* were calculated as the ratio between the minimum distance between the first and the fourth of 4 consecutive positions and the time interval between these positions. We choose to smooth the instant speed because time interval between 2 frames being very small (0.07 to 0.21s), displacement of 250nm corresponding to position error of diffracted limited microscopy resulted in high instant speed (1.2 to 3.6 µm/s) (S2F Fig). To compare the effect of treatments, statistical comparisons between median were performed in Prism software (Graphpad Inc.) as indicated in legends. All tests used in this report are two-sided. For visualization, the raw data were processed using ImageJ software (National Institutes of Health, Bethesda, MD, USA). Images stacks were processed as maximum projections and visualized after gaussian filter fixed at 0.5.

## Supporting information

**S1 Fig. a) Schematic representation of RSV-GFP-N infectious clone** (not to scale). Protein-encoding frames are shown as colored boxes, leader and trailer regions as hatched boxes. Intergenic regions are shown as black horizontal lines. The GFP-N coding sequence was inserted between M and SH genes together with an upstream gene end (GE) and a downstream gene

start (GS) signal as described in Methods. **b) Growth properties of RSV-GFP-N.** HEp-2 cells were infected with the RSV-GFP-N or RSV WT at a MOI of 1 at 37˚C and viruses were harvested at the indicated times p.i. and titrated by plaque assays on HEp-2 cells. Results are the mean ± s.d. for three independent experiments. Titers of RSV-GFP-N and RSV at the different time points are not significantly different using two-way ANOVA (*ns* for RSV versus RSV GFP-N; *ns* for interaction) followed by Sidak's multiple comparison test. **c) Colocalization of GFP-N with wild type N and P proteins in RSV-GFP-N infected cells.** HEp-2 cells were infected with RSV-GFP-N. At 24h p.i. cells were stained with antibodies against N (red) and P (cyan) and Hoechst 33342 (merge). The GFP-N protein is visualized through its spontaneous green fluorescence. RNPs are indicated with white arrow heads. Representative images are shown. Images were taken under a Leica SP8 confocal microscope. Images stacks (3 z-steps) were processed as maximum projections and visualized after gaussian filter fixed at 0.5. Scale bar 5 μm.
(TIF)

**S2 Fig. Illustration and schematic of tracking errors. a, b) Illustration of false tracking near the IBs. a)** Time projection of 14 consecutives frames over 1.95 s showing spot detections (grey spheres) and tracking (cyan dragon tails) by Imaris software on RSV-GFP-N infected HEp-2 as described in methods section. Artifactual tracks are visible near the large IB. Scale bar 5μm. A zoom of the region is shown on the right. **b)** Consecutives images of the area framed in orange (for clarity one out of two is removed). The top panel shows the original images, the lower panel shows the images with spots and track marks. Detected spots used to generate the tracks are pointed with white arrows on both panels. Note that no spots are visible on the original images. Tracks detected in the framed region on panel a) will be removed from the analysis. Scale bar 2μm **c, d, e) Illustration of false tracking due to wrong link between 2 slow moving objects. c)** Consecutive images showing spot detections (grey spheres) and tracking (yellow dragon tail) by Imaris software on RSV-GFP-N infected HEp-2 as described in methods section. In this example, the first detected particle is pointed by the yellow arrow and is animated with slow undirected motion. At time 0.75 s a new spot is detected (red arrow) and is associated with the track of the first one. This second spots exhibits also slow motion. Yellow and red arrows are pointing the same position on every images. Scale Bar 2μm. The spot is supposed to have covered approximately 0.8μm in 0.15 s resulting in an instant velocity of 5.3μm/s between these positions. This track will be filtered because all other instant velocities are below 2.6 μm/s (50% of the track maximum instant velocity). **d, e)** Schematic of the false tracking resulting from a wrong link between 2 slow moving spots. In each frame, previous position is indicated in fainted color and displacement is indicated by a black arrow. Scale bar 0.5μm. At time 0.75s the previous position of the first spot is linked to another particle position. Instant speed resulting from this schematic example are plotted in e. This track will be filtered as all instant speeds remain below 50% of the maximum instant speed of the track (green pointed line). **f)** Schematic representation of the consecutive real positions (circled letters) and detected positions (colored stars) of a slow-moving vRNP. Due to position errors of diffracted limited microscopy the calculated distance between the positions a and b could reach up to 200nm. As interval between 2 consecutive frames is 0.07s, the resulting calculated instant speed could reach up to 3μm/s while the particle remained quasi-immobile. To avoid this issue, we calculated *smooth instant speed* which is the ration between the minimal distance between the first and the fourth of 4 consecutive positions and the time interval. In this example, the distance between a and d is around 200nm and the time between frames is 0.28s, resulting in a smooth instant speed around 0.7μm/s.
(TIF)

**S3 Fig. Microtubules depolymerization impairs RSV growth.** A549 cells were infected with RSV at high MOI and Nocodazole 2μM (NZ) or DMSO was added 2 h post infection. At 24h viral titer of each sample was determined by plaque assay. Mean ± s.d. from 2 experiments in triplicate are shown. $^*$ $p$ <0.05 using Kolmogorov-Smirnov comparison test.
(TIF)

**S4 Fig. Colocalization of RNPs and Rab11a in RSV infected cells. a)** HEp-2 and A549 cells infected with RSV for 24 h. Rab11a (green) and RNPs (red) were detected by immunostaining. Representative images from one experiment are shown. Images stacks (3 z-steps) were processed as maximum projections and visualized after gaussian filter fixed at 0.5. Scale bar 5 μm. The boxed areas enclose Rab11 and N spots pointed by white arrows (zoom). **b)** Percentage of N spots colocalizing with Rab11a positive spots calculated using Icy Software. Results from individual cells from one experiment are plotted with mean and s.d. **c)** Extinction of Rab11a expression in SiRNA transfected cells. A549 cells were treated by siRNA Rab11a or non-targeting siRNA (control) for 48 h and 72h. Cell lysates were subjected to SDS-PAGE and probed by antibodies directed against Rab11a. The visualization of all proteins was realized by Stain Free revelation.
(TIF)

**S5 Fig. RSV infection increases Tf recycling.** HEp-2 cells infected with RSV GFP or mock infected for 24 hours were incubated with Alexa-647-Tf for 10min then washed and incubated in fresh medium for 0 to 20 min. Cells were fixed and observed under confocal microscopy. Total amount of Alexa-647-Tf per cell was quantified. Mean ± s.d. of relative amount of Alexa 647 Tf after 20 min of chase is shown. $^*$ $p$< 0.05 using t test with Welch's correction. Data are from 7 cells from one representative experiment out of 2.
(TIF)

**S1 Movie. RNPs dynamics in infected cells.** Time-lapse microscopy of RNPs in HEp-2 cells infected with RSV-GFP-N for 18–20 h.p.i. Cells were imaged as described in methods section every 0.21s for 1 min with a Spinning disk confocal X1. The resulting videos were visualized under Image J (6 f.p.s) after maximal z-projection of 3Z. A representative video from 2 independent experiments is shown. Scale bar 5 μm. Arrowheads point a RNP moving out of an IB.
(AVI)

**S2 Movie. Analysis of RNPs tracks.** Automatic tracking of moving RNPs was performed using Imaris software as described in methods section. The resulting videos were visualized under Imaris software. Results of tracking on time lapse images shown in S1 movie are presented here. Tracks are materialized by dragon tails. Scale bar of 5 μm.
(MP4)

**S3 Movie. RNPs moving along microtubules.** Time lapse microscopy of A549 cells infected with RSV-GFP-N for 18–20 h.p.i and treated with docetaxel-647. Cells were imaged as described in methods section every 0.15s for 1 min with a Spinning disk confocal X1. The resulting videos were visualized under Image J (8 f.p.s) after maximal z-projection of 2Z. A section of a representative video of 3 independent experiments is shown. Scale bar of 5 μm. The green arrow points a RNP moving along microtubules.
(AVI)

**S4 Movie. RNPs movements stop following microtubules depolymerization.** Time lapse microscopy of A549 cells infected with RSV-GFP-N for 18h. Images were taken every 0.63s in a $CO_2$-controlled chamber heated at 37˚C, with an Olympus FV3000 inverted confocal microscope. Cells were first imaged for 3 min, prior any drug treatment (left). Cells were then treated

with nocodazole (NZ) 20μM, and images were taken for another 3 min (right). Resulting videos were visualized using Image J software (8 f.p.s) after maximal z-projection of 3Z. A representative video from one experiment in which 3 videos were acquired is shown. Scale bar 5 μm.
(AVI)

**S5 Movie. Actin depolymerization doesn't stop RNPs movements.** Time lapse microscopy of A549 cells infected with RSV-GFP-N for 18h. Images were taken every 0.53s in a $CO_2$-controlled chamber heated at 37˚C, with an Olympus FV3000 inverted confocal microscope. Cells were first imaged for 3 min, prior any drug treatment (left). Cells were then treated with Cytochalasine D (CytoD) 2μM for 15 min, and images were taken for another 3 min (right). Resulting videos were visualized using Image J software (8 f.p.s) after maximal z-projection of 3Z. A representative video from one experiment in which 3 videos were acquired is shown. Scale bar 5 μm.
(AVI)

**S6 Movie. RNPs trafficking is not impaired by control DMSO treatment.** Time lapse microscopy of A549 cells infected with RSV-GFP-N for 18–20 h.p.i then treated with DMSO (control) for one hour. Cells were imaged as described in methods section every 0.15s for 1 min with a Spinning disk confocal X1. The resulting videos were visualized under Image J (12 f.p.s) after maximal z-projection of 2Z. A representative video of 3 experiments is shown. Scale bar 5μm.
(AVI)

**S7 Movie. Alteration of RNPs trafficking by microtubules depolymerization.** Time lapse microscopy of A549 cells infected with RSV-GFP-N for 18–20 h.p.i then treated with Nocodazole 2μM for one hour. Cells were imaged as described in methods section every 0.15s for 1 min with a Spinning disk confocal X1. The resulting videos were visualized under Image J (12 f.p.s) after maximal z-projection of 2Z. A representative video of 3 experiments is shown. Scale bar 5μm.
(AVI)

**S8 Movie. Dynamics of RNPs and Rab11a vesicles.** Time lapse microscopy of A549 cells transiently expressing mCherry-Rab11a and infected with RSV-GFP-N for 18–20 h.p.i. Cells were imaged as described in methods section every 0.07s for 1 min with a Spinning disk confocal X1. A representative video of 3 experiments is shown. The resulting videos were visualized under Image J (8 f.p.s). Scale Bar 5μm. The white arrows are pointing RNPs (green) moving together with Rab11a (white).
(AVI)

**S9 Movie. Alteration of RNPs associated with Rab11a mobility by microtubules depolymerization.** Time lapse microscopy of A549 cells transiently expressing Cherry-Rab11a and infected with RSV-GFP-N for 18–20 h.p.i. then treated for 10 min with nocodazole 20μM. Cells were imaged as described in methods section every 0.07s for 1 min with a Spinning disk confocal X1. A representative video of 2 experiments is shown. The resulting videos were visualized under Image J (8 f.p.s). Scale Bar 5μm. The white arrows are pointing RNPs (green) associated with Rab11a (red).
(AVI)

**S10 Movie. RNPs trafficking is not impaired by non targeted RNA silencing.** A549 cells were transfected with non targeting siRNA (control) for 48h and then infected with RSV-GFP-N. At 18–20 h.p.i., cells were imaged as described in methods section every 0.15s for

1 min with a Spinning disk confocal X1. The resulting videos were visualized under Image J (12 f.p.s) after maximal z-projection of 2Z. A representative video of 3 experiments is shown. Scale Bar 5μm.
(AVI)

**S11 Movie. Alteration of RNPs trafficking by RNA silencing of Rab11a.** A549 cells were transfected with non targeting SiRNA targeting Rab11a for 48h and then infected with RSV-GFP-N. At 18–20 h.p.i., cells were imaged as described in methods section every 0.15s for 1 min with a Spinning disk confocal X1. The resulting videos were visualized under Image J (12 f.p.s) after maximal z-projection of 2Z. A representative video of 3 experiments is shown. Scale Bar 5μm.
(AVI)

## Acknowledgments

We thank Marie Galloux, Jean-Francois Eléouët, Nadia Naffakh and Cédric Delevoye for sharing protocols and reagents, and for helpful discussions. We are grateful to Nathalie Sauvonnet for Rab11 plasmids and helpful discussions. We thank Jennifer Risso-Ballester, Cédric Diot and Sabine Blouquit-Laye for insightful discussions and critical reading of the manuscript. We thank Aude Jobart-Malfait and Cymages platform for access to the Leica SP8 microscope and Olympus FV3000 inverted confocal microscope, which was supported by grants from the region Ile-de-France. We also acknowledge the ImagoSeine core facility of the Institut Jacques Monod and thank Xavier Baudin for support for spinning disk imaging.

## Author Contributions

**Conceptualization:** Marie-Anne Rameix-Welti.

**Data curation:** Gina Cosentino.

**Formal analysis:** Gina Cosentino, Marie-Anne Rameix-Welti.

**Funding acquisition:** Marie-Anne Rameix-Welti.

**Investigation:** Gina Cosentino, Katherine Marougka, Aurore Desquesnes.

**Methodology:** Gina Cosentino, Marie-Anne Rameix-Welti.

**Software:** Nicolas Welti.

**Supervision:** Elyanne Gault, Marie-Anne Rameix-Welti.

**Visualization:** Gina Cosentino, Delphine Sitterlin, Marie-Anne Rameix-Welti.

**Writing – original draft:** Marie-Anne Rameix-Welti.

**Writing – review & editing:** Gina Cosentino, Katherine Marougka, Delphine Sitterlin, Elyanne Gault, Marie-Anne Rameix-Welti.

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
