## [Decision Letter · Decision Letter 0]

24 Apr 2022

Dear Dr. Rameix-Welti, 

Thank you very much for submitting your manuscript "Respiratory syncytial virus ribonucleoproteins hijack microtubule Rab11 dependent transport for intracellular trafficking" for consideration at PLOS Pathogens. As with all papers reviewed by the journal, your manuscript was reviewed by members of the editorial board and by several independent reviewers. The reviewers appreciated the attention to an important topic. However, both of them identified a number of points which will need to be addressed. In light of the reviews, we would like to invite the resubmission of a revised version that takes into account the reviewers' comments.

Sincerely,

Jianrong Li

Guest Editor

PLOS Pathogens

Guangxiang Luo

Section Editor

PLOS Pathogens

Kasturi Haldar

Editor-in-Chief

PLOS Pathogens

orcid.org/0000-0001-5065-158X

Michael Malim

Editor-in-Chief

PLOS Pathogens

orcid.org/0000-0002-7699-2064

Reviewer Comments (if any, and for reference):

Reviewer's Responses to Questions

**Part I - Summary**

Reviewer #1: The authors have generated a recombinant RSV with a second copy of the N (nucleocapsid) protein with GFP fused to its N-terminus, inserted further down the genome so that it would be produced in lower amounts than the unmodified N. This modified RSV replicates as well as its parent and can be visualized in infected cells in culture. They have found that Rab11b co-localizes with the nucleocapsid and is co-transported along microtubules within live cells. Nocodazole, a disruptor of microtubules, inhibits, but does not completely halt this transport. The results illustrate rapid transport of RSV nucleocapsids through cells using the recycling endosome mechanism.

Reviewer #2: Cosentino et al present a well-written and well-executed paper that demonstrates that RSV utilizes microtubules and Rab-11 for transport of vRNPs within the cytoplasm. A method was developed to track vRNPs in live cells. Measurements to track speed and distance are necessarily somewhat arbitrary but are sufficient to support the main findings. The movies are both supportive of their claims and beautiful to watch. Previous papers have demonstrated the importance of microtubules for RSV replication and the effects of nocodazole, but the present work makes a first step to begin to understand the mechanism. Some of the bigger questions also posed by the authors, how and whether vRNPs move from factory sites to the membrane, and the role of inclusion bodies therein, remains unexplored and undiscussed. This somewhat lessens the impact of the paper, as it not surprising that RSV, like many other negative-strand RNA viruses, uses microtubules and rab11.

**Part II – Major Issues: Key Experiments Required for Acceptance**

Reviewer #1: The experiments are very thorough, with multiple approaches and assays to demonstrate this interaction, transport and quantify its speed in several ways. The results are clear and convincing.

Reviewer #2: Whereas the demonstration of vRNP movement within the cytoplasm and involvement of Rab11 is a step forward in understanding, the role of inclusion bodies therein (which are by many believed to replication factories) and where vRNPs are formed and migrate to, in preparation for assembly, is the more exciting and relevant question. Do fast-moving vRNPs represent a major mechanism underlying assembly or artefacts? Experimentally, this may be the next dimension of study, however any discussion on how the data fits within the current understanding of assembly steps, and involvement of inclusion bodies, is lacking.

Several older papers, such as Kallewaard listed by the authors but also several other papers, have found that in addition to microtubules, cortical F-actin also plays a role in the late stages of assembly. These findings should be discussed in the paper.

Although GFP-N is incorporated into vRNPs, we cannot know whether GFP affects N-coating of vRNPs or vRNP movement. For example, it cannot be excluded that GFP slows down vRNP movement, and that faster motion is under-represented. This does not diminish the findings but should be acknowledged.

Fig 7. In this paragraph, please indicate more clearly in the text which part of the figure is referred to.

Fig. 7b. There seems to be only a very small amount of P, especially given the high amounts of HA-Rab11a and N on the same blot. This finding does not support that P is co-precipitated, contrary to the statement in line 307.

**Part III – Minor Issues: Editorial and Data Presentation Modifications**

Reviewer #1: A few grammatical changes are suggested on the attached manuscript copy.

Reviewer #2: - line 67: add reference

- lines 275 and 276: define ER. Do the authors mean RE?

- Fig 6: define the difference between * and **.

- line 337-338. Recent work by Piedra et al suggests the transcription gradient (and protein levels) may be more complex than anticipated. Hence, it cannot be predicted whether N-GFP levels would be lower than N levels.

- line 381: first use of ERC should be defined.

PLOS authors have the option to publish the peer review history of their article (what does this mean?). If published, this will include your full peer review and any attached files.

Reviewer #1: No

Reviewer #2: No

Figure Files:

Data Requirements:

Reproducibility:

References:

---

## [Editor Report · Decision Letter 1]

25 May 2022

Dear Dr. Rameix-Welti,

We are pleased to inform you that your manuscript 'Respiratory syncytial virus ribonucleoproteins hijack microtubule Rab11 dependent transport for intracellular trafficking' has been provisionally accepted for publication in PLOS Pathogens.

Best regards,

Jianrong Li

Guest Editor

PLOS Pathogens

Guangxiang Luo

Section Editor

PLOS Pathogens

Kasturi Haldar

Editor-in-Chief

PLOS Pathogens

orcid.org/0000-0001-5065-158X

Michael Malim

Editor-in-Chief

PLOS Pathogens

orcid.org/0000-0002-7699-2064

The authors have adequately addressed the reviewers' comments.
---

## [Editor Report · Acceptance letter]

16 Jun 2022

Dear Pr Rameix-Welti,

We are delighted to inform you that your manuscript, "Respiratory syncytial virus ribonucleoproteins hijack microtubule Rab11 dependent transport for intracellular trafficking," has been formally accepted for publication in PLOS Pathogens.

Best regards,

Kasturi Haldar

Editor-in-Chief

PLOS Pathogens

orcid.org/0000-0001-5065-158X

Michael Malim

Editor-in-Chief

PLOS Pathogens

orcid.org/0000-0002-7699-2064